# Development of a new tool for the assessment of patient-defined benefit in hospitalised older patients: the Patient Benefit Assessment Scale for Hospitalised Older Patients (P-BAS HOP)

Maria Johanna van der Kluit [1], Geke J Dijkstra,[2] Barbara C van Munster,[1,3] Sophia De Rooij[1,4]

**To cite:** van der Kluit MJ, Dijkstra GJ, van Munster BC, et al. Development of a new tool for the assessment of patient-defined benefit in hospitalised older patients: the Patient Benefit Assessment Scale for Hospitalised Older Patients (P-BAS HOP). *BMJ Open* 2020;**10**:e038203. doi:10.1136/bmjopen-2020-038203

► Prepublication history and additional materials for this paper is available online. To view these files, please visit the journal online (http://dx.doi.org/10.1136/bmjopen-2020-038203).

For numbered affiliations see end of article.

**Correspondence to**
Maria Johanna van der Kluit;
m.j.van.der.kluit@umcg.nl

## ABSTRACT

**Objectives** To support the shift from disease-oriented towards goal-oriented care, we aimed to develop a tool which is capable both to identify priorities of an individual older hospitalised patient and to measure the outcomes relevant to him.

**Design** Mixed-methods design with open interviews, three step test interviews (TSTIs) and a quantitative field test.

**Setting** University teaching hospital and a regional teaching hospital.

**Participants** Hospitalised patients ages 70 years and older.

**Results** The Patient Benefit Assessment Scale for Hospitalised Older Patients (P-BAS HOP) consists of a baseline questionnaire and an evaluation questionnaire. Items were based on 15 qualitative interviews with hospitalised older patients. Feedback from a panel of four community-dwelling older persons resulted in some adaptations to wording and one additional item. Twenty-six hospitalised older patients participated in TSTIs with Version 1 of the baseline questionnaire, revealing indications for a good content validity and barriers in completion behaviour, global understanding and understanding of individual items, which were solved with several adaptations. Four additions were made by participants. After TSTIs with ten patients with the evaluation questionnaire, one adaptation was made. A field test with 91 hospitalised older patients revealed a small number of missing values.
To enhance the feasibility, the number of items was reduced from 32 to 22, based on correlations and mean impact score. The field test was repeated with 104 other patients in a regional teaching hospital. To enhance the understanding, the tool was split into two phases. This version was tested with TSTIs with eight patients and appeared to be understandable. The final version was an interview-based tool and took about 11 min to complete.

**Conclusions** The P-BAS HOP is a potentially suitable tool to identify priorities and relevant outcomes of the individual patient. Further research is needed to investigate its validity, reliability and responsiveness.

### Strengths and limitations of this study

► The content of the Patient Benefit Assessment Scale for Hospitalised Older Patients (P-BAS HOP) is based on open interviews with hospitalised older patients.
► Patients are able to indicate their individual outcome priorities.
► The P-BAS HOP is tested intensively in the target population with three-step test interviews. This gave valuable insights into the understanding of the tool and the completion behaviour of the participants.
► The current version of the P-BAS HOP is only suitable to be completed with an interviewer and not as a self-administered questionnaire.
► It is unknown whether the P-BAS HOP is feasible in other healthcare systems, languages and cultures than in the Netherlands.

## BACKGROUND

To fit the needs of the ageing population, and patients with multiple chronic diseases, a shift is recommended from disease-oriented towards goal-oriented care. Older patients with multimorbidity may be more interested in more personal goals such as for them important symptoms, functional status and social functioning than in traditional outcomes such as survival and biomarkers,[1 2] but these goals and outcomes differ per individual.[3 4] When care would be systematically evaluated by personal goal-oriented outcomes, a tool is needed which is capable both to identify the priorities of the individual patient and to measure the outcomes relevant to him.

Three literature reviews[5–7] into tools used to assess patient outcome priorities in the context of multimorbidity revealed a few

BMJ

potentially useful tools. Tools only suitable for specific activities, such as the Canadian Occupational Performance Measure (COPM),[5 7] Self-Identified Goals Assessment (SIGA),[7] Assessment of Motor and Process Skills (AMPS) and McMaster Toronto Arthritis (MACTAR)[5] were ignored. A general tool is the Outcome Prioritisation Instrument,[6 8] which is suitable to elicit four patient priorities, but these priorities are still very global and it remains unclear how to evaluate them after treatment. Another tool is the Goal Attainment Scaling (GAS),[5 7] which is designed to set and evaluate individualised goals and outcomes. Disadvantages of the GAS are that it can be too challenging for patients to articulate their own goals and that it is time consuming.[9] The International Classification of Functioning, Disability and Health (ICF) framework for goal setting is used to categorise patient goals set in semistructured interviews, but has still the same disadvantages as the GAS and has, in addition, a very poor responsiveness.[7] Finally, with the Target Complaints,[5] the patient defines target complaints as those problems for which help was sought. These complaints are scored at the beginning and at the end of the treatment by the patient, or the patient rates the degree of improvement.[10 11] The Target Complaints is individualised and patient centred. However, it focuses solely on problems and not on goals. Disadvantages for the GAS, ICF and Target Complaints could be that for some older patients it might be difficult to formulate their own goals and problems because many older persons are not accustomed to defining and discussing personal goals and prompting is often necessary.[12] The quality of the answers is therefore dependent on the interviewer's experiences and techniques.

For this reason, another method of defining patient-defined goals and outcomes was sought and found in the literature about treatment of acne. Augustin *et al*[13] developed a tool consisting of two parts: (1) a baseline questionnaire to assess the importance of various predefined goals, based on themes derived from qualitative interviews in patients with acne and (2) an evaluation questionnaire to evaluate the extent to which treatment helped to achieve these goals. Based on these data, it is possible to compute an individual Patient Benefit Index. This is an overall value between 0 (no benefit) and 4 (maximal benefit), which reflects the achievement of the goals weighted by the importance.[13] The advantage of this tool is the insight into the individualised patient perspective, together with standardisation.

The aim of this study was to develop a tool to inventory individual goals and benefits of older hospitalised patients, based on the model of Augustin *et al*.[13] This article presents its development, early testing and adaptations.

## METHODS
The steps used to develop the Patient Benefit Assessment Scale for Hospitalised Older Patients (P-BAS HOP) are based on the steps of De Vet *et al*[14] and outlined in figure 1. After each step, the tool was adapted. The steps

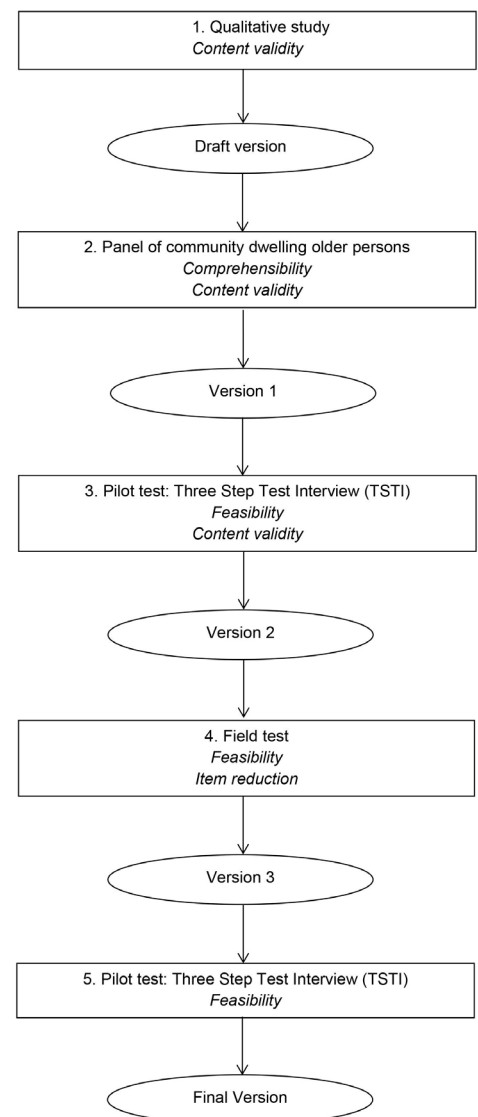

**Figure 1** Development of the Patient Benefit Assessment Scale for Hospitalised Older Patients (P-BAS HOP).

are explained in the following sections. For the readability, the methods and results of each step are alternated. The P-BAS was developed and tested in Dutch. The P-BAS was translated into English in a translation—back translation procedure involving four translators (two native English, two native Dutch), a language professional and authors MJvdK and GJD.[15]

### Patient and public involvement
Patients and public were involved in the generation of the items, the importance and relevance of the items and the assessment of the feasibility and understanding of the tool.

### Qualitative study
First, open interviews with hospitalised older medical and surgical patients about their goals regarding their hospitalisation were performed. The description of these goals is published elsewhere.[3] These goals were then coded inductively and transformed into questionnaire items,

and the first draft of the P-BAS HOP was then constructed, consisting of a baseline questionnaire and an evaluation questionnaire.

## Panel of community-dwelling older persons

The first draft version was proposed by email to a panel of community-dwelling older persons to assess the comprehensibility and relevance of the items and the tool and ask for omissions or redundancies.

### Results

Four community-dwelling older persons gave written feedback on the draft version of the tool. This led to adaptation in wording of the introduction text and to some items which were not clear enough or could be interpreted in multiple ways. An example of an item that was adapted: for the item 'to be able to eat', it was not clear if it concerned the instrumental activity of eating or concerned appetite. Therefore, the item was changed into 'can take pleasure in eating'.

Further, one additional item was added, namely 'to wash and dress yourself' and the sequence of two items was changed. Version 1 of the P-BAS HOP is shown in online supplemental appendix 1.

### Pilot test: three step test interview

The adapted tool (Version 1, online supplemental appendix 1) was tested with the three-step test interview (TSTI)[16 17] in older hospitalised patients. The TSTI is a type of cognitive interview suitable to assess how people interpret a questionnaire, its different items and what kind of strategies they use in responding to them. The TSTI consists of the following steps:

#### Step 1: concurrent thinking aloud

The participant completed Version 1 of the P-BAS HOP while thinking aloud. The interviewer observed, made notes of the participant's behaviour (hesitations, skipping questions, corrections) and verbalised thoughts. However, the interviewer did not talk, or intervene. The instructions for the participant were: Please fill in this questionnaire and try to think aloud about what your thoughts are while reading the questions and choosing the right response category.

#### Step 2: retrospective interview

With the retrospective interview any gaps from the first step were filled in. Every behaviour and thought from the observation of which the interviewer wanted further information, was clarified.

#### Step 3: semistructured interview

An in-depth interview was conducted, aimed at eliciting the participant's considerations and opinions. The participant was given the opportunity to explain behaviour, actions or thoughts that he had in the previous steps. The participant was asked how he understood different items, was asked for any omissions and his opinion about filling in the questionnaire. The participant was also asked to

explain his goals in his own words in order to perform a first content validation of the P-BAS HOP.

### Participants

Eligible participants of the TSTI were 70 years and older; planned or unplanned hospitalised on medical or surgical wards of a university teaching hospital in the Netherlands; able to speak and understand Dutch and were without cognitive impairment. Inclusion criteria were verified with the staff nurse, and patients were then approached by the interviewer (MJvdK). Participants were completely anonymous, no list with names or other identifying data was made, nor did the researchers have access to medical records. Participants gave verbal consent to the interview and audio recording.

### Data analysis

Data gathering and data analysis were alternated. Interviews were audio recorded and transcribed verbatim. All remarks were then organised by question and step. After that, the data were coded by MJvdK and grouped into categories. The tool was adapted several times after the feedback until it was considered feasible and understandable.

The TSTI was repeated with the evaluation questionnaire. This was done at patient discharge.

### Results

#### Sample baseline questionnaire

Twenty-six older hospitalised patients participated in the TSTI. Characteristics of the participants are displayed in the second column of table 1.

#### Coding categories

The codes were sorted into the following categories: completion behaviour, global understanding, understanding and reactions regarding individual items, additions, general evaluation and content validity.

#### Completion behaviour

With 'completion behaviour' is meant the behaviour participants showed when completing the questionnaire. Observations revealed that many participants skipped the instruction text partly or even completely. For some participants, reading these instructions was demanding, others did not understand how a table works and where to place a mark. Adding an example table with instructions showing how and where to place a mark brought no improvements. In the final version boxes to tick were included in the table.

#### Global understanding

Many examples of correct understanding were coded. For example:

> Item regaining weight, step 1: Oops, dear, pooh, let's see, that is certainly important, because I have lost weight lately since I had not been feeling well for a while, that is, not shortness of breath or anything, but, very tired, listless. So, a little weight gain is important. (P9)

**Table 1** Participants three steps test interview baseline questionnaire Version 1, evaluation questionnaire Version 1 and baseline questionnaire Version 3

| | Baseline questionnaire Version 1 (n=26) | Evaluation questionnaire Version 1 (n=10) | Baseline questionnaire Version 3 (n=8) |
|---|---|---|---|
| Characteristic | n | n | n |
| Gender | | | |
| Male | 19 | 8 | 4 |
| Female | 7 | 2 | 4 |
| Age (years) | | | |
| 70–79 | 18 | 8 | 7 |
| 80–89 | 7 | 2 | 0 |
| 90–99 | 1 | 0 | 1 |
| Native language | | | |
| Dutch | 14 | 7 | 7 |
| Local dialect | 10 | 2 | 1 |
| Frisian | 2 | 0 | 0 |
| Foreign language | 0 | 1 | 0 |
| Educational level* | | | |
| Low | 8 | 5 | 2 |
| Middle | 10 | 4 | 5 |
| High | 8 | 1 | 1 |
| Admission reason† | | | |
| Cardiac problems | 9 | 1 | 1 |
| Pulmonary problems | 7 | 3 | 2 |
| Bowel problems | 2 | 2 | 1 |
| Fever/infection | 2 | 2 | 0 |
| Vascular surgery | 2 | 1 | 0 |
| Cancer | 2 | 0 | 0 |
| Accident/fracture(s) | 2 | 0 | 1 |
| Kidney problems | 0 | 1 | 1 |
| Syncope | 0 | 0 | 1 |
| Ulcera | 0 | 0 | 1 |

*Definition educational level: low=no education, primary school, basic vocational training; middle=secondary education, vocational training; high=bachelor, master.
†Reason according to the patient.

Or:

Item walking, step 1: Well, I walk well. Doesn't apply to me. (P17)

A few participants interpreted the questions as if it was an evaluation of their current level of functioning. For example:

Item energy, step 2: I: You have filled in 'not at all' in 'you have more energy'. What is the reason that you just …? P: Because I feel lethargic. That is what I mean to say. I used to be a very energetic person. (…) That is gone. (…) That is what I mean by that question. I: Yes, so you actually filled in how you are feeling now. P: Yes, now. At the time. I: Yes, so you say… P: Not from last year or half a year ago. They are snapshots, aren't they? That was what you meant, right? (…) I: And when I ask you the question: 'How important is it that you get energy again?' P: Very important. (…) Because I've always been energetic. Very important. (P2)

Other participants had difficulties relating the goals to their own situation. For example:

Item shortness of breath, step 1 I'm actually never short of breath. But it is quite important. (P25)

There were also participants who did recognise that a certain goal did not apply for them, but they did not understand how to indicate that in the tool.

van der Kluit MJ, *et al*. *BMJ Open* 2020;**10**:e038203. doi:10.1136/bmjopen-2020-038203

Some participants were reluctant to use the options 'not at all important' or 'doesn't apply to me', because they deemed those answers socially undesirable.

Since many older persons have multiple health problems, it is possible that a participant experiences a problem with an item, but is admitted for another health problem. Many participants were able to make this distinction. For example:

> Item moving, step 1: Yes, that will never be all right again, I can tell. Does no longer apply at all. Already 30 years ago they said: Mr. B., you have to learn to live with that. And they still say that today. Osteoarthritis, there is nothing you can do about it. (P18)

But for others this distinction was more difficult.

### Adaptations
To enhance the general understanding, the following adaptations to the tool were made and tested in new participants:

Several adaptations were made in the instruction text.

In the columns with the answer options the word 'important' was added to all answer options. For example: 'very' was changed into 'very important', to make clear that the question was not to evaluate current function, but to indicate how important the goal was.

The sequence of the questions was changed. To enable participants to relate the goals to their own situation, the questions related to somatic complaints were moved to the beginning of the questionnaire.

Another adaptation made to improve the understanding to relate the goals to the patients' own situation, was to add the word 'again' to the goals, to make clear that it is something they had before and they have to regain by the hospital admission. For example: How important is it to you that you have normal bowel movements again.

The next adaptation was to move the answer option 'doesn't apply to me' from the last to the first column. This made it easier to find that option.

The final adaptation to improve making the connection between the hospital admission and their goals, was repeating the question in every line. Instead of having the text 'How important is it to you that by this hospital admission…' on top of the page alone, this question was repeated in every row.

Apart from this, several adaptations were made to the layout in order to ease the reading for participants.

### Understanding and reactions regarding individual items
The following individual items caused discussion: take pleasure in eating, to know the cause of your complaints, take a short break and remain alive.

### Take pleasure in eating
Some participants had a more epicurean association with this item. Therefore, it was changed into: 'regain appetite'.

### Cause of complaints
With the item 'how important is it for you that you know the cause of your complaints?' some participants spontaneously started to describe risk factors like smoking. By changing the item into 'knowing what is wrong with you', this was solved.

### Take a short break
The item 'can take a short break' gave many different interpretations, often without any relationship with the hospital. Several alternatives were tried: 'to recharge', 'to take a moment', but these did not improve the understanding. It was, therefore, decided to remove this item.

### Remain alive
The item 'remain alive' gave mixed reactions. Some were irritated by the question. For others it was obvious that it was very important to them that they wanted to remain alive, by adding words like 'of course!'. However, there were also participants who deemed remaining alive less obvious and started to think about the question. Unless the mixed reactions to this question, it was remained because it was not obvious for all participants and because the researchers considered it unreasonable to have a questionnaire with many potential outcomes, but to omit the one outcome that for many participants is considered as the most important.

### Additions
Participants gave the following suggestions which were added to the questionnaire: family life, driving, hobbies, urinating. The adaptations and additions led to P-BAS HOP Version 2.

### General evaluation
Many participants stated that the questionnaire was quite easy to fill out, although this was not always congruent with the observations about their understanding. Several mentioned enjoying filling in the questionnaire. One participant mentioned that the tool was very important for him in order to state his own priorities. For another participant, the questionnaire was considered emotional, because the questions were confronting and he was afraid that many goals were not feasible. For some the questionnaire was somewhat tiring.

### Content validity
The goals the participants mentioned in their own words, were qualified in the questionnaire as at least 'somewhat important' in almost all cases. For example:

> Yes, that is the quality of life… Yes, it is important that comes up to standard again. (…) Well, cycling that, that comes in second place. I think walking is more important than ehm… (…) I have been a volunteer for more than forty years now, helping people fill out tax forms. I think that is important to me. And that is, that is, that is also the volunteer work. If it is somewhat possible I would like to do that again. (…) Ehm,

go on outings. I would like to keep doing things like that. (P14)

This participant filled in in the questionnaire: Walking: quite important, (volunteer) work: moderately important, go on outings: moderately important.

### TSTI evaluation questionnaire
#### Sample evaluation questionnaire
Ten patients participated in the TSTI for the evaluation questionnaire at discharge. The sampling of the participants continued until the last version of the questionnaire was considered clear and did not reveal any new problems. Characteristics of the participants are displayed in the third column of table 1.

#### Process of testing and adaptations
The process of testing and adapting the evaluation questionnaire was much faster, because many problems with layout and wording of individual items had already been solved in the baseline phase. In the first version, the wording appeared to be too complicated for some participants. Therefore, the original formulation: 'The hospitalisation helped me to….' Was changed into: 'Because of the hospitalisation….'. This adaptation was clear for all the following participants and led to Version 2.

### Field test with version 2: item reduction based on mean impact score and correlation
Version 2 was tested with a new group of hospitalised older patients. The aim of this field test was to assess the feasibility of the P-BAS HOP in combination with other questionnaires. The trained research assistants observed during the field test that the tool was too time consuming and that some patients still had difficulties relating the questions to their personal situation, as was observed in the TSTI. Therefore, the following extra adaptations were made: item reduction, answer option reduction, and splitting the tool into two phases.

### Participants
Eligible participants were consecutive patients aged 70 years and older; planned or unplanned hospitalised on medical or surgical wards of a university teaching hospital, expected to stay for at least 48 hours; and at maximal 4 days hospitalised at the moment of interviewing; able to speak and understand Dutch and were without cognitive impairment. Inclusion criteria were verified with the staff nurse. Patients were approached by a trained research assistant and gave signed informed consent to participate. The questionnaire was then conducted in a face-to-face interview with the research assistant, but to patients in a better condition and with middle or higher education the opportunity was given to fill in the questionnaire themselves, an option which only a minority of patients choose.

### Item reduction
As this is a formative tool, item reduction procedures suitable for reflective tools, such as based on factor analysis and Cronbach's alpha, are not relevant.[14] Item reduction was therefore based on correlation and mean impact score.

Items within one category with a strong correlation, measured probably the same construct. Therefore, from dyads with a Spearman's rank order correlation >0.7, one item was removed.[14] For the calculation of the Spearman's rank order correlation coefficient, the answer option 'does not apply to me now' and 'not at all important' were coded as 0, the options somewhat, moderately, quite important and very important were coded, respectively, as 1–4.

For the reduction based on mean impact score, all items were sorted into categories. For each item the mean impact score was calculated: [% for whom the item applied] × [mean importance for that item]. From every category with two or more items, the item with the lowest mean impact score was removed.[14] The field test was repeated in a regional teaching hospital by a trained research assistant, to check whether the impact differed in another context.

### Results
The Benefit Assessment Scale Version 2 consisted of 32 items. In the 3-month inclusion period, 492 consecutive eligible patients meeting the inclusion criteria were admitted on the selected wards. Of these patients, 238 were not approached for logistic reasons, for example the patient could not be interviewed within the first 4 days because of absence for treatment, transfer from ward, shortage of research assistants. Hence, 254 patients were approached for informed consent and 106 patients (42%) gave informed consent. Of the 106 included patients, the P-BAS was not administered 15 times because of lack of time (eg, patient had to leave for treatment or discharge) or the patient was too tired. This resulted in 91 administered P-BAS questionnaires. Of the 91 participants, 20 answered the questionnaire independently written and 71 were interviewed by the research assistant. Characteristics of the participants are displayed in table 2 and the results are shown in table 3.

As seen in table 3, the number of missing values ranges from 0 to 4 per item. The answer options with the lowest priorities were used the least, especially 'not at all important' and 'somewhat important'. Therefore, and also because on reflection the options 'somewhat' and 'moderately' were very close, we decided to remove the option 'moderately'.

Four dyads had a Spearman's rank-order correlation coefficient >0.7: energy and condition ($r_s$=0.80); moving and walking ($r_s$=0.87); cooking and groceries ($r_s$=0.75); cooking and housekeeping ($r_s$=0.70) Therefore of these dyads, one item was removed (condition, moving and cooking), inspired by the information derived from the TSTIs.

Table 4 shows the items with mean impact scores, sorted per category and descending mean impact scores. From the categories with at least two items, the item with the

Table 2 Participants field test (n=91)

| Characteristic | n |
| --- | --- |
| Gender | |
| Male | 63 |
| Female | 28 |
| Age (years), median (range) | 75 (70–96) |
| Native language | |
| Dutch | 55 |
| Local dialect | 27 |
| Frisian | 3 |
| Unknown | 6 |
| Educational level* | |
| Low | 22 |
| Middle | 47 |
| High | 22 |
| Specialty | |
| Medical | 42 |
| Surgical | 23 |
| Cardiology | 26 |
| Admission type | |
| Acute | 60 |
| Elective | 31 |

*Definition educational level: low=no education, primary school, basic vocational training; middle=secondary education, vocational training; high=bachelor, master.

lowest mean impact score was removed. To give participants still the opportunity to indicate their individual priorities, even when being a minority, we added an open option to add extra individual goals.

## Repetition field test in regional teaching hospital

The field test was repeated in a regional teaching hospital with the same items, but with fewer answer options and the questions in two steps, as explained in the next paragraph. In the 8-week inclusion period, 209 patients meeting the inclusion criteria were admitted on the wards. Of these patients, 56 were not approached for logistic reasons. A total of 153 were, therefore, approached for informed consent and 104 patients (67%) gave informed consent. The items with the lowest mean impact scores were the same for most categories, except for the categories independence/freedom, improving daily functioning and work/hobbies.

## Splitting tool into two phases

Since some problems with understanding remained, especially the difficulties relating the goals to their own situation, as described in the TSTI, we decided to split the tool into two phases. In the first phase an inventory of subjects with problems or limitations was made. These could be problems/limitations at the moment of interview, at admission, or expected problems/limitations. In

the second phase, only the importance was asked for the goals related to the subjects that applied. As this adaptation complicated the tool, we decided to use it as an interview-based tool. The item reduction and splitting into two phases, resulted in P-BAS HOP Version 3.

### TSTI with version 3

Version 3 was tested again with the TSTI in hospitalised older patients. The procedure was identical as in step 3, however, as this version is only applicable as an interview version, this was done with an interviewer and observant. The observant only observed during the first step, and took over the interview role in the second and third steps.

### Results

Eight participants participated in the TSTI about Version 3. Characteristics of the participants are displayed in the last column of table 1.

#### *General understanding*

In general, the tool in two phases was well understood. For example:

> Item shortness of breath, phase 1, step 1: No, I have no problems with that, you know, shortness of breath. (A1)

Or:

> Item shortness of breath, phase 1, step 1: Yes, that is present! And for that reason, I am admitted here. My oxygen was too low. And my carbon dioxide level is not good, much too high. Yes, complication of, yes. (A3)

We shortened the instructions, but did not modify the content of the tool. This last adaptation led to the final questionnaire (online supplemental appendix 2). The completion of this baseline questionnaire took 5–24 min, with a median of 11 min.

## DISCUSSION

The P-BAS HOP was constructed as a tool that should be capable both to identify the goals and priorities of the individual older hospitalised patient and to measure the outcomes relevant to him regarding hospitalisation.

The items of the P-BAS HOP were based on interviews with hospitalised older patients. Including patients in the generation of patient reported outcomes is not self-evident and is even absent in many cases.[18] But even when patients are involved in the generation of outcomes, they still only reflect the priorities of the overall patient population and not the individual patient. Therefore, the major advantage of the P-BAS HOP is that patients can indicate their individual priorities, which also leads to individual benefit scores.

Indicating individual priorities is also possible with the GAS, but the GAS is more time-consuming, varying from 15 to 20 min for experienced assessors,[19] to 90 minutes per patient,[20] while the P-BAS HOP takes 5–24 min, with

**Table 3** Scores of Version 2 benefit assessment scale baseline (n=91)

| Item | Missing Failed* n (%) | n.d.† n (%) | Does not apply to me now n (%) | Importance Not at all n (%) | Some-what n (%) | Moderately n (%) | Quite n (%) | Very n (%) |
|---|---|---|---|---|---|---|---|---|
| Better | 1 (1.1) | 0 | 8 (8.8) | 0 | 1 (1.1) | 2 (2.2) | 17 (18.7) | 62 (68.1) |
| Weight | 1 (1.1) | 1 (1.1) | 57 (62.6) | 10 (11.0) | 4 (4.4) | 7 (7.7) | 3 (3.3) | 8 (8.8) |
| Condition | 0 | 0 | 17 (18.7) | 2 (2.2) | 1 (1.1) | 6 (6.6) | 31 (34.1) | 34 (37.4) |
| Energy | 0 | 1 (1.1) | 18 (19.8) | 1 (1.1) | 0 | 3 (3.3) | 33 (36.3) | 35 (38.5) |
| Pain | 1 (1.1) | 1 (1.1) | 33 (36.3) | 0 | 2 (2.2) | 4 (4.4) | 9 (9.9) | 41 (45.1) |
| Bowel movements | 0 | 1 (1.1) | 58 (63.7) | 4 (4.4) | 2 (2.2) | 1 (1.1) | 12 (13.2) | 13 (14.3) |
| Urinate | 0 | 1 (1.1) | 64 (70.3) | 4 (4.6) | 0 | 1 (1.1) | 10 (11.0) | 11 (12.1) |
| Shortness of breath | 1 (1.1) | 0 | 39 (42.9) | 1 (1.1) | 2 (2.3) | 5 (5.5) | 11 (12.1) | 32 (35.2) |
| Walking | 0 | 1 (1.1) | 32 (35.2) | 1 (1.1) | 3 (3.3) | 5 (5.5) | 16 (17.6) | 33 (36.3) |
| Moving | 0 | 1 (1.1) | 35 (38.5) | 1 (1.1) | 2 (2.2) | 5 (5.5) | 18 (19.8) | 29 (31.9) |
| Appetite | 0 | 2 (2.2) | 55 (60.4) | 1 (1.1) | 2 (2.2) | 6 (6.6) | 9 (9.9) | 16 (17.6) |
| Knowing what is wrong | 0 | 1 (1.1) | 32 (35.2) | 2 (2.2) | 2 (2.2) | 1 (1.1) | 12 (13.2) | 41 (45.1) |
| Disease under control | 0 | 1 (1.1) | 10 (11.0) | 0 | 0 | 2 (2.2) | 15 (16.5) | 63 (69.2) |
| Alive | 0 | 1 (1.1) | 2 (2.2) | 2 (2.2) | 1 (1.1) | 1 (1.1) | 13 (14.3) | 71 (78.0) |
| Enjoy | 0 | 2 (2.2) | 20 (22.0) | 0 | 0 | 2 (2.2) | 13 (14.3) | 54 (59.3) |
| Freedom | 0 | 1 (1.1) | 31 (34.1) | 0 | 1 (1.1) | 1 (1.1) | 12 (13.2) | 45 (49.5) |
| Cooking | 0 | 2 (2.3) | 51 (56.0) | 1 (1.1) | 4 (4.4) | 2 (2.2) | 14 (15.4) | 17 (18.7) |
| Housework | 0 | 1 (1.1) | 51 (56.0) | 2 (2.2) | 5 (5.5) | 5 (5.5) | 10 (11.0) | 17 (18.7) |
| Groceries | 0 | 1 (1.1) | 42 (46.2) | 1 (1.1) | 5 (5.5) | 9 (9.9) | 12 (13.2) | 21 (23.1) |
| Wash and dress | 0 | 2 (2.3) | 51 (56.0) | 0 | 0 | 2 (2.2) | 14 (15.4) | 22 (24.2) |
| Garden | 0 | 1 (1.1) | 48 (52.7) | 2 (2.2) | 1 (1.1) | 8 (8.8) | 10 (11.0) | 21 (23.1) |
| Sports | 0 | 1 (1.1) | 46 (50.5) | 7 (7.7) | 2 (2.2) | 9 (9.9) | 7 (7.7) | 19 (20.9) |
| Hobbies | 0 | 1 (1.1) | 39 (42.9) | 1 (1.1) | 2 (2.2) | 4 (4.4) | 13 (14.3) | 31 (34.1) |
| Work | 0 | 3 (3.3) | 63 (69.2) | 3 (3.3) | 0 | 4 (4.4) | 7 (7.7) | 11 (12.1) |
| Driving | 0 | 2 (2.2) | 46 (50.2) | 1 (1.1) | 0 | 2 (2.2) | 8 (8.8) | 32 (35.2) |
| Outings | 0 | 1 (1.1) | 28 (30.8) | 1 (1.1) | 3 (3.3) | 8 (8.8) | 19 (20.9) | 31 (34.1) |
| Visiting | 0 | 3 (3.3) | 28 (30.8) | 1 (1.1) | 4 (4.4) | 6 (6.6) | 19 (20.9) | 30 (33.0) |
| Family life | 0 | 4 (4.4) | 39 (42.9) | 0 | 0 | 2 (2.2) | 12 (13.2) | 34 (37.4) |
| Home | 0 | 2 (2.3) | 21 (23.1) | 1 (1.1) | 0 | 1 (1.1) | 4 (4.4) | 62 (68.1) |
| Independence | 1 (1.1) | 3 (3.3) | 29 (31.9) | 0 | 1 (1.1) | 2 (2.2) | 6 (6.6) | 49 (53.8) |

*Measurement failed: invalid answer due to two options filled in.
†No answer was given.
n.d., not done.

a median of 11 min. Moreover, for some older patients it might be difficult to formulate their own goals,[12] and the P-BAS HOP helps patients with examples of predefined goals.

More recently, models for goal based decision making were developed,[21–23] but these methods are more suitable for clinical encounters to align treatment option with patient goals. The major advantage of the P-BAS HOP is that it is a more suitable and efficient tool to measure personalised outcomes in, for example, trials. It also could replace a diversity of existing tools, since it covers several dimension like symptoms, daily functioning, social functioning. Examples for which the P-BAS HOP could be used are to compare the personalised outcomes for alternatives of hospital admission, such as,[24–27] the effectiveness of better geriatric management of in-hospital patients,[28] or in a narrower way, to compare the effectiveness of different treatment methods on personalised outcomes.

The pilot and field tests of the P-BAS HOP started already before we achieved complete saturation of goals in the qualitative interviews. Therefore, patients had the possibility to add goals during the TSTI. Several goals were added during the TSTI, which also appeared later in the qualitative interviews.[3] Still, the qualitative interviews revealed later some extra target complaints, which were

**Table 4** Mean impact scores per category

| Goals | University hospital | | | Regional teaching hospital | | |
|---|---|---|---|---|---|---|
| | Applied (%) | Importance score (M) | Mean impact score | Applied (%) | Importance score (M) | Mean impact score |
| **Remain alive** | | | | | | |
| Remain alive | 98 | 3.7 | 3.62 | 75 | 2.64 | 1.9 |
| **Controlling disease** | | | | | | |
| Controlling disease | 89 | 3.76 | 3.34 | 29 | 2.43 | 0.7 |
| **Improving condition** | | | | | | |
| Feeling better | 91 | 3.71 | 3.38 | 71 | 2.73 | 1.94 |
| Energy | 80 | 3.4 | 2.72 | 50 | 2.23 | 1.12 |
| *Condition* | 81 | 3.27 | 2.66 | 65 | 2.34 | 1.53 |
| *Weight* | 36 | 1.84 | 0.66 | 9 | 2.33 | 0.2 |
| **Alleviating complaints** | | | | | | |
| Pain | 63 | 3.59 | 2.26 | 44 | 2.72 | 1.2 |
| Breath | 57 | 3.39 | 1.92 | 38 | 2.64 | 0.99 |
| Appetite | 38 | 3.09 | 1.18 | 35 | 2.39 | 0.83 |
| Bowel | 35 | 2.88 | 1.03 | 29 | 2.47 | 0.71 |
| *Urinate* | 29 | 2.92 | 0.83 | 17 | 2.67 | 0.46 |
| **Enjoying life** | | | | | | |
| Enjoying life | 78 | 3.75 | 2.91 | 31 | 2.53 | 0.78 |
| **Improving/maintaining social functioning** | | | | | | |
| Outing | 69 | 3.23 | 2.23 | 27 | 2.11 | 0.57 |
| Visiting | 68 | 3.22 | 2.2 | 21 | 1.91 | 0.4 |
| *Family life* | 55 | 3.67 | 2.03 | 5 | 2.8 | 0.13 |
| **Knowing what is wrong** | | | | | | |
| Wrong | 64 | 3.52 | 2.27 | 39 | 2.58 | 0.99 |
| **Regaining/maintaining independence, freedom** | | | | | | |
| Home | 76 | 3.85 | 2.94 | 15 | 2.5 | 0.39 |
| Independence | 67 | 3.78 | 2.52 | 17 | 2.44 | 0.42 |
| *Freedom* | 66 | 3.71 | 2.43 | 23 | 2.54 | 0.59 |
| **Improving daily functioning** | | | | | | |
| Walking | 64 | 3.33 | 2.14 | 54 | 2.57 | 1.38 |
| *Moving* | 61 | 3.31 | 2.02 | 34 | 2.54 | 0.86 |
| Driving | 48 | 3.63 | 1.75 | 14 | 2.13 | 0.31 |
| Groceries | 53 | 2.98 | 1.59 | 19 | 2.3 | 0.44 |
| Wash/dress | 43 | 3.53 | 1.52 | 26 | 2.52 | 0.65 |
| *Cooking* | 43 | 3.11 | 1.33 | 15 | 1.75 | 0.27 |
| *Housework* | *43* | *2.9* | *1.26* | *20* | *1.86* | *0.38* |
| **Resuming work/hobbies** | | | | | | |
| Hobbies | 57 | 3.39 | 1.92 | 20 | 2 | 0.4 |
| Garden | 47 | 3.12 | 1.46 | 16 | 1.24 | 0.2 |
| Sports | 49 | 2.66 | 1.3 | 23 | 1.58 | 0.37 |
| *Work* | *28* | *2.92* | *0.83* | *13* | *1.92* | *0.24* |

The removed items are indicated in italic.

not included in the P-BAS HOP, such as vomiting, dizziness and sweating. Yet, in the final version of the P-BAS HOP, patients still have the opportunity to add personal goals which were not mentioned before.

By using the mean impact score to reduce items, items considered least important by the overall sample were removed, though this does not take account of the priorities of individuals who deviate from the majority. For this

reason the extra open option was added. Most removed items, based on mean impact score, were confirmed when repeated in the regional teaching hospital. The only exceptions were in the categories improving daily functioning, resuming work/hobbies and regaining/maintaining independence/freedom.

In the categories improving daily functioning and resuming work/hobbies, the lowest priorities were 'housework' and 'work' in the first sample and 'driving' and 'gardening' in the second. Since driving and work were the second lowest priority in the second sample, the removal of housework and work could be justified.

In the category regaining/maintaining independence/freedom priorities in both hospitals were entirely opposite. We, therefore, have to conclude that we were too early to remove the item freedom. It is unclear whether these differences are caused by different contexts or because the field test in the regional hospital was after splitting the questionnaire into two phases, and therefore, the questions were altered.

## LIMITATIONS

The P-BAS HOP is only tested in hospitalised patients without cognitive impairment. It is therefore unknown if it is suitable in other contexts and it might be too complex for patients with cognitive impairment. In addition, the P-BAS HOP is only tested in the Netherlands and the translated English version has not yet been tested. Therefore, it is unknown whether the P-BAS HOP is applicable in other languages and cultures.

The TSTI gave valuable insights into the understanding of the questionnaire and the completion behaviour of the participants. Many adaptations were made, but it proved difficult to make the questionnaire understandable for all patients. These kinds of difficulties were seen in various examples where the TSTI was used.[16 29–31] Unfortunately, the final version is only suitable to be completed with an interviewer and not as a self-administered questionnaire. The TSTI gave a first indication of the content validity, but further quantitative research into the construct validity, in which the priority of goals can be compared with experienced symptoms or limitations at admission and the achievement of goals can be compared with progression or deterioration of other constructs, test–retest reliability of baseline and evaluation questionnaire and responsivity to test the validity of the PBI is needed.

## CONCLUSIONS

The P-BAS HOP is a potentially suitable interview-based tool to identify the priorities and relevant outcomes of the individual older hospitalised adult. Based on these data, it is possible to compute an individual Patient Benefit Index, which is an overall value between 0 (no benefit) and 3 (maximal benefit), which reflects the achievement of the goals weighted by the importance. Further quantitative research is needed to investigate the construct validity, reliability and responsiveness.

**Author affiliations**
[1]University Center for Geriatric Medicine, University Medical Centre Groningen, University of Groningen, Groningen, The Netherlands
[2]Department of Health Sciences, Applied Health Research, University Medical Center Groningen, University of Groningen, Groningen, The Netherlands
[3]Department of Geriatrics, Gelre Hospitals, Apeldoorn, The Netherlands
[4]Medical School Twente, Medical Spectrum Twente, Enschede, The Netherlands

**Acknowledgements** We would like to thank all patients who participated in the interviews for sharing their stories with us and their feedback on the different versions of the P-BAS HOP. We thank the panel members for their feedback on the draft version. We would also like to thank all the research assistants and students for the many interviews they conducted, especially Gerdine Boot for all the patients she interviewed in the regional teaching hospital. We would like to thank Daniël Bosold, Janet Vroomen, Caroline Mutsaers, Margaretha Schenkeveld and Anna Pot for their role in the translation process of the P-BAS HOP.

**Contributors** MJvdK designed the study. MJvdK conducted the three-step test interviews (TSTIs). MJvdK coded the data of the TSTIs. MJvdK and SDR regularly discussed the codes and the progress and adaptations of the different versions of the P-BAS HOP. SDR supervised the field test in the University hospital, BCvM in the Regional teaching hospital. MJvdK wrote the first draft of the manuscript, GJD, BCvM and SDR contributed significantly to subsequent manuscript revisions. All authors have read and approved the final version of the manuscript.

**Funding** This study was funded by an unrestricted grant from the Dutch Foundation for Effective Elderly Care and the University of Groningen.

**Competing interests** None declared.

**Patient consent for publication** Not required.

**Ethics approval** The Medical Ethics Research Committee of the UMCG (file number M16.192615 and M16.199647) confirmed that the Medical Research Involving Human Subjects Act did not apply to the research project. Official approval by the committee was, therefore, not required.

**Provenance and peer review** Not commissioned; externally peer reviewed.

**Data availability statement** All data relevant to the study are included in the article or uploaded as online supplemental information.

**ORCID iD**
Maria Johanna van der Kluit http://orcid.org/0000-0001-8758-232X

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
