## [Reviewer comments · BMJ Open]

ARTICLE DETAILS

TITLE (PROVISIONAL)	Development of a new tool for the assessment of patient defined benefit in hospitalised older patients: the Patient Benefit Assessment Scale for Hospitalised Older Patients (P-BAS HOP)
AUTHORS	van der Kluit, Maria ; Dijkstra, Geke; van Munster, Barbara; De Rooij, Sophia

VERSION 1 – REVIEW

REVIEWER	Louise Hickman UTS Australia
REVIEW RETURNED	25-Apr-2020

GENERAL COMMENTS	Authors state official approval ethics by the committee was not required - why? The reviewer provided a marked copy with additional comments. Please contact the publisher for full details.
---

REVIEWER	Aanand D. Naik Baylor College of Medicine, USA
REVIEW RETURNED	08-May-2020

GENERAL COMMENTS	Thank you for the opportunity to review this article. The authors are addressing an important and understudied area of research. Valid and reproducible methods for conducting and measuring goal-oriented care are needed urgently. The authors should be commended for taking on a rigorous instrument development process with multiple levels of participant input and testing. Furthermore, the investigators have chosen to enroll a complex, difficult to recruit population of complex adults during hospital admissions. These elements of the study design are rigorous and difficult to accomplish. Despite my support for the significance and importance of this research area and need for instrument development, there are significant limitations to the study design and analysis that temper my enthusiasm for this manuscript. 1) Introduction. The authors have not fully described the literature on goal-directed approaches to care, especially among older and complex patients. Recent work by Elywn and Vermunt, Hargraves and Montori, and Tinetti and Naik provide examples of approaches to goals based care and efforts to elicit goals among patients as a part of routine care planning and outcome assessment. In particular, the recent study by Tinetti et al. 2019 in JAMA Internal Medicine, and the description of their priorities identification process previously published in J Am Geriatrics Soc 2018 (Naik et al.). 2) Methods. The TSTI methodology is well described and
--

	implemented. However, there is not an otherwise clear proposed methodology for item validation and reliability testing, construct validation or rationale for conducting or not conducting factor analysis. 3) Item Reduction. The methodology for conducting item reduction is not well described or justified. In particular, why is mean impact score and correlation a more appropriate or stronger approach compared with other approaches? How is the impact score correlations used to remove items – could an item of high conceptual importance be removed due to a low impact score? 4) Similarly, the rationale and approach for answer option reduction is not adequately described. 5) The rationale for why the investigators split the instrument into two phases needs to be strengthened. There should be some theoretical or conceptual basis for how the two phases were designed and then some empirical rules or principles guiding how the split occurred. Neither seem well described. 6) Overall, the process of instrument development and validation needs a stronger conceptual overview and guidance over the course of the methods and results. Some of the decisions are not clear. I suggest using one or another of the frameworks for survey development and validation. 7) Discussion. The discussion section could be strengthened by explaining why and how understanding of goals and priorities could improve the care and/or care outcomes of hospitalized adults with complex illnesses. 8) How does the approach used by the investigators differ conceptually (not just practically) from Goal Attainment Scaling? Is their approach stronger conceptually compared to GAS? 9) Some comment in the discussion section on the practical challenges and barriers to using this new scale during routine clinical encounters. How would the authors envision a clinician using the survey during routine care? 10) The limitations section is not clearly labelled.
--	---

VERSION 1 – AUTHOR RESPONSE

Reviewer: 1

Reviewer Name: Louise Hickman

Institution and Country: UTS Australia

Please state any competing interests or state ‘None declared’: None

Please leave your comments for the authors below

Authors state official approval ethics by the committee was not required - why?

Response: We did present our study to the Medical Ethics Research Committee of the UMCG. This committee confirmed that the Medical Research Involving Human Subjects Act (WMO) did not apply to the research project. Research is subject to the WMO if the following criteria are met: 1.It concerns medical scientific research and 2.Participants are subject to procedures or are required to follow rules of behaviour

(<https://english.ccmo.nl/>). Therefore the Medical Ethics Research Committee declared it did not have to give formal approval.

Remarks copied from attached file:

Abstract:

Over all it would be good to edit/synthesis the abstract whilst providing specific numbers and details for the readers.

Response: As you suggested, we added numbers in the abstract.

Objective: would be good to defend why an existing tool would not be suitable.

Response: We agree that this would be a good idea, but since the abstract has a word limit of 300 words, we do not have the opportunity in the abstract. However, we did so in the introduction.

‘... which is capable both to identify priorities of an individual older hospitalised patient and to measure his relevant outcomes’.

‘his’ needs editing ‘him’.

Response: We edited the sentence.

Results: ‘based on qualitative interviews with hospitalised older patients’ how many was this 4?

Response: 15 interviews. We added this information in the abstract.

TSTIs: Keep in full avoid acroynms.

Response: we agree that it is clearer to write acronyms out in the abstract, but ‘Three Step Test Interviews’ are too many (four!) words to repeat several times in the abstract. We therefore decided to write it out the first time and use the acronym for the succeeding situations.

‘TSTIs with Version 1 of the baseline questionnaire with twenty-six hospitalised older patients showed (...) After TSTIs with ten patients with the evaluation questionnaire, one adaptation was made.’

with 26 patients then with an additional 10 - this is not clear to the reader.

Response: 26 participants participated in the TSTI with the baseline questionnaire and 10 participants participated in the TSTI with the evaluation questionnaire. We modified the text to make this more clear to the reader.

‘the number of items was reduced...’ what was or is the number of items?

Response: We reduced the number of items from 32 to 23 items. We added this information in the abstract.

‘The field test was repeated with 104 patients in a regional teaching hospital’. 104 different patients from the previous 94?

Response: Correct. We made this more clear in the text.

‘Conclusions: The P-BAS HOP is a potentially suitable tool to identify priorities and relevant outcomes of the individual patient’.

What are these relevant outcomes and priorities the tool may assess well? are they supported priorities from other research in this area?

Response: This is an interesting question, however we think this is not the place to discuss the comparison with other literature. Firstly, since this is an abstract, we don’t have room for discussion here because of the word limit. Secondly, this article is about the development and testing of a tool. The content of the tool (items) is discussed in another article, namely the qualitative interviews we discussed in the first step. In that article we compared the outcomes with other literature. This article is about the instrument. The uniqueness of the P-BAS HOP is that patient chooses the items that are relevant to him and does not have to evaluate items that are not important or relevant to him.

Background

IADLs or ADL tools are internationally used please describe why these were not appropriate given their wide spread use.

Response: If we were sure that all patients had the goal improving or maintaining (I)ADL, we would have used one of these instruments. However, goals are unique for every single person. That was

the reason we searched for a tool that is capable to identify the priorities of the individual patient and to measure only outcomes relevant to that person. We added some extra argumentation in the first paragraph.

‘a tool is needed which is capable both to identify the priorities of the individual patient and to measure **his** relevant outcomes.’ ??

Response: We edited the sentence into ‘to measure the outcomes relevant to him’.

Methods

Figure one helps - but in the current form the methods is very difficult to follow. See if there is another way to present this mixed methods approach so it is clear to the reader.

Response: We can imagine that this iterative process may be difficult to follow by the reader. We therefore decided to maintain Figure 1 as an outline of the process used, but to alternate the methods and results, to guide the reader step by step.

Qualitative study - my understanding is this is published elsewhere?

Response: Yes, this is well understood.

‘Eligible participants of the TSTI were 70 years and older; planned or unplanned hospitalised on medical or surgical wards of a university teaching hospital in the Netherlands;’ one hospital setting?

Response: Yes, this was in one hospital setting.

Results

TSTI - Can the author provide a shorter synthesis of the next four pages please?

Response: We shortened the text with 367 words, mainly by shortening or omitting citations.

Field test with Version 2. – ‘Of these patients, 238 were not approached for **logistic reasons**’. what were these?

Response: We added some examples of logistic reasons in the text. Examples were: patient could not be interviewed within the first four days because of absence for treatment, transfer from ward, shortage of research assistants

Discussion

Could it be shorter and used alongside existing tools within the health system?

Response: We already shortened the tool for feasibility reasons. We think that a further shortening would diminish the multidimensionality of the tool. It could be used alongside existing tools within the health system, but it also could replace existing tools. As the P-BAS is a personalised tool, it could replace a test set of ADL, symptom, and social functioning instruments. We added this suggestion in the Discussion.

‘Therefore, the major advantage of the P-BAS HOP is that patients can indicate their individual priorities, which also leads to individual benefits.’

Can the authors link this statement to specific supportive results?

Response: We think this sentence was not clear to the reader. What we meant to say was that the P-BAS HOP leads to individual benefit-scores expressed in the Patient Benefit Index. We therefore changed the sentence.

Did the group complete cognitive screening prior?

Response: As described in the methods-sections, cognitive impairment was an exclusion criterion. This was not assessed by formal cognitive screening, but verified with the staff nurse.

There are many tools available for clinicians for certain areas identified within the P-BAS HOP. Can the authors please place the results in context for clinicians - such as can this tool replace existing tools with better outcomes?

Response: This tool could potentially replace existing tools, since it measures personalised goals and outcomes. Therefore it could replace existing test sets of ADL, symptom, social functioning instruments, with the advantage that it only measures outcomes relevant for that individual. But we think it is too early to conclude that the P-BAS can replace existing tools with better outcomes,

because we do not have information yet about the reliability and validity. However, as mentioned above, we did a suggestion in the Discussion.

Does this tool need further work and validation?

Response: Yes, as mentioned in the Discussion and Conclusions, further quantitative research is needed to investigate the construct validity, reliability and responsiveness.

Due to the iterative mixed methods design the methods and results are difficult to follow in parts. Please adjust where possible for ease of translation to the reader.

Response: We can imagine that this iterative process may be difficult to follow by the reader. We therefore decided to maintain Figure 1 as an outline of the process used, but to alternate the methods and results, to guide the reader step by step.

Reviewer: 2

Reviewer Name: Aanand D. Naik

Institution and Country: Baylor College of Medicine, USA

Please state any competing interests or state 'None declared': None Declared

Please leave your comments for the authors below

Thank you for the opportunity to review this article.

The authors are addressing an important and understudied area of research. Valid and reproducible methods for conducting and measuring goal-oriented care are needed urgently. The authors should be commended for taking on a rigorous instrument development process with multiple levels of participant input and testing. Furthermore, the investigators have chosen to enroll a complex, difficult to recruit population of complex adults during hospital admissions. These elements of the study design are rigorous and difficult to accomplish.

Despite my support for the significance and importance of this research area and need for instrument development, there are significant limitations to the study design and analysis that temper my enthusiasm for this manuscript.

1) Introduction. The authors have not fully described the literature on goal-directed approaches to care, especially among older and complex patients. Recent work by Elywn and Vermunt, Hargraves

and Montori, and Tinetti and Naik provide examples of approaches to goals based care and efforts to elicit goals among patients as a part of routine care planning and outcome assessment. In particular, the recent study by Tinetti et al. 2019 in JAMA Internal Medicine, and the description of their priorities identification process previously published in J Am Geriatrics Soc 2018 (Naik et al.).

Response: We are familiar with the mentioned literature and highly value the contribution made to goal based care and decision making. As model for conversation and care alignment, we think the models you mentioned are superior to the P-BAS HOP. However we think the P-BAS HOP is a more suitable and efficient tool to measure personalised outcomes in, for example, trials. In the introduction we made an overview of potential instruments for goal identification and outcome measurement and not of models for conversation and care alignment. However, we agree that it is very worthwhile to discuss the models you proposed and compare them with the P-BAS HOP. We added therefore the information in the Discussion.

2) Methods. The TSTI methodology is well described and implemented. However, there is not an otherwise clear proposed methodology for item validation and reliability testing, construct validation or rationale for conducting or not conducting factor analysis.

Response: We agree that further validation and reliability testing of the instrument is very important, however this is too much information for one article. We therefore did a proposal in the Discussion and Conclusions for further quantitative research into the construct validity, reliability and responsiveness. See next paragraph for rationale for not using factor analysis.

3) Item Reduction. The methodology for conducting item reduction is not well described or justified. In particular, why is mean impact score and correlation a more appropriate or stronger approach compared with other approaches? How is the impact score correlations used to remove items – could an item of high conceptual importance be removed due to a low impact score?

Response: In multi-item measurement instruments, two models can be distinguished: reflective models and formative models.¹ In a reflective model, the construct is reflected by the items and the items correlate with each other and may be interchangeable. In a formative model, the items form or cause the construct and do not necessarily correlate with each other. The P-BAS is a formative instrument, and this has consequences for the method of item reduction. For formative instruments, frequently used methods for item reduction as factor analysis and Cronbach's alpha are not suitable.

For that reason, the mean impact score is most useful for item reduction in formative scales.¹ By using the pre-specified categories, we maintained the multi-dimensionality of the instrument. The removed items were not considered relevant or important by a majority of the participants. The items with a strong correlation were considered interchangeable.

As we mentioned in the Discussion-section, removing items based on a low mean impact score, has a disadvantage, namely that an item could still be very relevant for a small minority of the participants. We therefore included an 'open option'.

To make this more clear, we moved some justification for the method of item reduction from the Discussion- section to the Methods-section.

4) Similarly, the rationale and approach for answer option reduction is not adequately described.

Response: As described in the section 'Field test with version 2', we had the following arguments for item reduction: the instrument was too time consuming, the answer options with low priorities were used rarely, and 'somewhat important' and 'moderately important' are on reflection very close. Therefore we decided to remove the answer option 'moderately'. This resulted in the logical sequence of 'not at all', 'somewhat', 'quite' and 'very' important.

5) The rationale for why the investigators split the instrument into two phases needs to be strengthened. There should be some theoretical or conceptual basis for how the two phases were designed and then some empirical rules or principles guiding how the split occurred. Neither seem well described.

Response: the rationale for splitting the tool into two phases was more empirical than theoretical. As described in the pilot test with the TSTI, some participants had difficulties to relate the goals to their own situation. These participants rated goals as important while these were not applicable to their personal situation. These problems still appeared during the field test. We therefore sought for a method to make it easier for the participant to relate the goals to his own situation and to precede the questions about the importance of the goal with a step to sort out which subjects are applicable to the person. This method is subsequently again empirically tested in a new TSTI.

6) Overall, the process of instrument development and validation needs a stronger conceptual overview and guidance over the course of the methods and results. Some of the decisions are not clear. I suggest using one or another of the frameworks for survey development and validation.

Response: As described in the Methods section, we used the framework for instrument development by De Vet et al,¹ consisting of construct definition, item development, pilot testing and field testing. Since each step is followed by an evaluation, leading to adaptation, and therefore again evaluation, this process is iterative and not straight forward. We therefore decided to maintain Figure 1 as an outline of the process used, but to alternate the methods and results, to guide the reader step by step.

7) Discussion. The discussion section could be strengthened by explaining why and how understanding of goals and priorities could improve the care and/or care outcomes of hospitalized adults with complex illnesses.

Response: We added some information about the models for goal based decision making, which makes clear that understanding goals can help aligning treatment options with patient goals and so can inform decision making.

8) How does the approach used by the investigators differ conceptually (not just practically) from Goal Attainment Scaling? Is their approach stronger conceptually compared to GAS?

Response: Apart from the time aspect, the P-BAS HOP differs from the GAS that it contains examples of predefined goals, while with the GAS, the patient has to articulate his own goals, which can be challenging for many older people. We added this information in the Discussion.

9) Some comment in the discussion section on the practical challenges and barriers to using this new scale during routine clinical encounters. How would the authors envision a clinician using the survey during routine care?

Response: We have no experience yet with the use of this tool in routine clinical encounters. However, we added some argumentation that the P-BAS HOP could replace a diversity of existing tools.

10) The limitations section is not clearly labelled.

Response: we added a heading 'Limitations'.

VERSION 2 – REVIEW

REVIEWER	Aanand Naik Baylor College of Medicine, Houston, Texas USA
REVIEW RETURNED	23-Jul-2020

GENERAL COMMENTS	Thank you for the opportunity to review this manuscript. The authors are commended for their comprehensive and thorough response to prior reviews. The revised manuscript is very complete and goes through the development and validation/revision process of the instrument clearly. The manuscript will serve as a helpful reference and tool for describing the initial development/validation of the instrument to guide subsequent testing and use. The only additional comment/suggestion I will make is for the discussion section. In contrasting this tool with other more clinical approaches, the authors make reference to its use primarily as a research tool. I would ask that they expand on this a bit further. Provide suggestions for 1) how/when it can be used; 2) what are the research gaps it can fill; 3) what specifically are examples of further validation studies they are undertaking or suggest that others do. Thank you for this important contribution to the field and literature!
--